# Stable Luminescent Poly(Allylaminehydrochloride)-Templated Copper Nanoclusters for Selectively Turn-Off Sensing of Deferasirox in β-Thalassemia Plasma

**DOI:** 10.3390/ph14121314

**Published:** 2021-12-16

**Authors:** Hung-Ju Lin, Chun-Chi Wang, Hwang-Shang Kou, Cheng-Wei Cheng, Shou-Mei Wu

**Affiliations:** 1School of Pharmacy, College of Pharmacy, Kaohsiung Medical University, Kaohsiung 807, Taiwan; Weiting0825@gmail.com (H.-J.L.); kouhs@kmu.edu.tw (H.-S.K.); pharmacysniksam@yahoo.com.tw (C.-W.C.); 2Department of Medical Research, Kaohsiung Medical University Hospital, Kaohsiung 807, Taiwan; 3Drug Development and Value Creation Research Center, Kaohsiung Medical University Hospital, Kaohsiung 807, Taiwan; 4Department of Fragrance and Cosmetic Science, College of Pharmacy, Kaohsiung Medical University, Kaohsiung 807, Taiwan; 5Taiwan Food and Drug Administration, Ministry of Health and Welfare, Taipei 11561, Taiwan

**Keywords:** PAH-Cu NCs, sensing, deferasirox, DFX, β-thalassemia, plasma

## Abstract

Highly stable and facile one-pot copper nanoclusters (Cu NCs) coated with poly(allylamine hydrochloride) (PAH) have been synthesized for selectively sensing deferasirox (DFX) in β-thalassemia plasma. DFX is an important drug used for treating iron overloading in β-thalassemia, but needs to be monitored due to certain toxicity. In this study, the PAH-Cu NCs showed highly stable fluorescence with emission wavelengths at 450 nm. The DFX specifically interacted with the copper nanocluster to turn off the fluorescence of the PAH-Cu NCs, and could be selectively quantified through the fluorescence quenching effect. The linear range of DFX in plasma analyzed by PAH-Cu NCs was 1.0–100.0 µg/mL (r = 0.985). The relative standard deviation (RSD) and relative error (RE) were lower than 6.51% and 7.57%, respectively, showing excellent reproducibility of PAH-Cu NCs for sensing DFX in plasma. This method was also successfully applied for an analysis of three clinical plasma samples from β-thalassemia patients taking DFX. The data presented high similarity with that obtained through a capillary electrophoresis method. According to the results, the PAH-Cu NCs could be used as a tool for clinically sensing DFX in human plasma for clinical surveys.

## 1. Introduction

Recently, sensing technology for fast detection has become a major field in detection strategies, such as the detection of COVID-19. Most studies have utilized nanomaterials that have been widely developed in the past decade as the detection system [1,2]. Among the metal nanomaterials, copper nanomaterials are cheaper and more stable than gold or silver. Therefore, recently, several studies have developed methods for synthesizing different types of copper nanoclusters (Cu NCs), including template-based [3], electrochemical [4], microemulsion [5], microwave [6], and modified Brust–Schiffrin [7] methods. Cu NCs show good fluorescent properties, electronic properties, and utilization for catalytic activities [7]. Although Cu NCs exhibit many advantages, they are unstable without the protection of certain materials. For this reason, many studies have developed and used various materials to increase the stability of Cu NCs, such as protein [8,9,10], DNA [11,12], amino acid [13], polymer [14], or thiols [15,16,17] components. Here, we propose a one-pot method for synthesis of highly stable Cu NCs by using poly(allylamine hydrochloride) (PAH) as a protecting agent and L-ascrobic acid (LAA) as a reducing agent. In previous studies, PAH has been used as the reducing agent [18], protecting agent [19], and coating material [20,21] for nanoparticles, but never utilized for formation of Cu NCs. In this study, PAH-coated Cu NCs were synthesized and hypothesized to serve as a sensor for specific detection of an iron chelating agent, i.e., deferasirox (DFX and Exjade^®^), in β-thalassemia patients.

β-Thalassemia, a highly prevalent hereditary disease, is caused by the absence of beta-globin chains. Three types of β-thalassemia have been identified according to severity, including major, intermediate, and minor types [22]. Two major therapeutic strategies, i.e., bone marrow transplantation and long-term blood transfusion, have been proposed for treating β-thalassemia. Clinical rejection reaction often occurs in the transplantation of bone marrow, and thus blood transfusion is a safe therapy for treating β-thalassemia. However, iron ions can accumulate in the body due to a continuous blood transfusion and can induce syndromes such as hemochromatosis and peroxidative tissue damage [23,24,25] as a result of iron overloading. Iron overloading results in high mortality of blood-transfusion β-thalassemia patients without effectively controlling the level of iron. To prevent iron overloading, iron chelating agents are administrated to transfusion patients to remove excess iron from the human body.

Deferasirox (DFX, Exjade^®^), 4-[3,5-bis-(hydroxyphenyl)-1,2,4-triazol-1-yl]-benzoic acid, is a tridentate iron chelator (Appendix A) that forms a complex of two molecules of DFX with a ferric ion (Fe^3+^) in a neutral condition [26]. DFX is an oral iron chelator and was approved by FDA and EU in 2005 and 2006, respectively. It can eliminate overloaded iron ions by forming water-soluble iron-DFX excreted from feces [27,28,29]. In clinical practice, DFX has been introduced as the first-line therapy for β-thalassemia patients who have accepted blood transfusions over 2 years. Although DFX has been shown to be excellent for treating iron overloading, more attention should be given to some severe adverse effects, including gastrointestinal bleeding, acute liver necrosis [30], and renal dysfunction [30,31,32], especially in children. Due to these adverse effects, it is very important to establish an easy-operating method for monitoring the DFX concentration in a patient’s plasma. Until now, a lot of analytical methods have been developed for determining DFX in plasma, and most of them have utilized HPLC coupled with UV, fluorescence, or MS/MS [33,34,35,36,37]. Recently, a stacking CE technique [38] was proposed for measuring DFX in biological fluid. Although these techniques are robust, they require highly expensive instruments, complex operating procedures, and skills for separation. Sensing is a simple, convenient, and quick strategy for detecting certain analytes, and is widely applied in clinical practice. Wang et al. even utilized the dopamine-conjugated carbon dots for sensing DFX in plasma [39]. In this study, a facile and economical one-pot synthesis method was proposed for synthesizing PAH-coated Cu NCs to be used as a sensor for detection of DFX in plasma. The mechanism is shown in Figure 1. The specific binding of DFX with PAH-coated Cu NCs induced aggregation of PAH-coated Cu NCs, and then the fluorescence of the PAH-coated Cu NCs was decreased. Therefore, the PAH-coated Cu NCs could be used for specifically sensing DFX. The method was successfully applied for detecting DFX in the plasma of β-thalassemia patients and it is feasible that it could serve as a tool for routine determination of DFX in clinical practice.

## 2. Results and Discussion

### 2.1. Optimization of PAH-Cu NCs Synthesis

To obtain the optimal PAH-Cu NCs, the effect of each factor in the synthesis of PAH-Cu NCs was investigated. Reaction time and temperature are two factors that affect the synthesis efficiency of PAH-Cu NCs. The data (Figure 1A) indicated that the fluorescence of PAH-Cu NCs was stable during the reaction time from 2 to 7 h and at a reaction temperature of 60 °C. However, because different morphological nanomaterials were produced with increasing temperature [40], the fluorescent intensity declined dramatically at 70 °C and 80 °C, at 2 h. To obtain highly stable fluorescence of PAH-Cu NCs and to shorten the reaction time, the reaction time and temperature were set at 2 h and 60 °C, respectively.

Because the pH value of the reaction buffer would affect ionization and, furthermore, the fluorescent intensity of as-prepared PAH-Cu NCs, the range of pH value from 2.0 to 7.0 was explored. As shown in Figure 1B, the as-prepared PAH-Cu NCs had the strongest fluorescent intensity at a pH value of 3.0 with uniform particle size distribution. The particle size at pH values from 5.0 to 7.0 was larger or irregular as compared with that at pH 3.0. Park [41] described that PAH would become flocculation at higher pH values, and this phenomenon was also observed at pH values of 5.0, 6.0, and 7.0 in this study. Therefore, the PAH-Cu NCs solution was observed to become turbid at pH values from 5.0 to 7.0. Taking into consideration the stability and fluorescent intensity of PAH-Cu NCs, the as-prepared PAH-Cu NCs were synthesized at a pH value of 3.

In previous research, polymeric amine has been used as a reducing agent to synthesize nanoparticles [18]. Herein, PAH was utilized as the reducing agent and also the protecting agent for synthesis of Cu NCs. To obtain the highly stable and fluorescent PAH-Cu NCs, different amounts of PAH from 2% to 12% were evaluated (Figure 1C). The PAH-Cu NCs had the highest fluorescent intensity by using 10% PAH, where the particle size and shape were also regular and uniform. Therefore, the optimal condition for synthesizing PAH-Cu NCs was controlled by using 10% PAH.

LAA, a mild reducing agent, is frequently used as the reducing metal ion to form nanoparticles [42]. As compared with other reducing agents [43,44,45,46], LAA is environmentally friendly and easier for operation. The Cu^2+^ could be encapsulated by LAA and reduced into Cu(0). Because the size of Cu NCs is regulated by the concentration of LAA [47], the concentration of LAA was evaluated to obtain stable PAH-Cu NCs (Figure 1D). By increasing the LAA concentrations, the fluorescent intensity of the PAH-Cu NCs was increased. When the concentration of LAA reached 1.0 M, the PAH-Cu NCs had the highest fluorescence. However, the fluorescence of PAH-Cu NCs decreased by using higher than 1.0 M LAA, which supported the fact that a higher concentration of LAA induced smaller Cu NCs which were unstable under the protection of PAH. Therefore, the LAA concentration for synthesis of PAH-Cu NCs was set at 1.0 M.

### 2.2. Characterization of the PAH-Cu NCs

In a previous study [42], PAH was used as a protecting and reducing agent to form other metal nanoparticles. In this study, PAH was utilized as a protecting agent to form PAH-Cu NCs. The fluorescence of Cu NCs can be reduced completely within several minutes without adding PAH, which supported using PAH to coil the Cu NCs with presenting amine groups to increase the stability of Cu NCs. The amount of PAH was optimized, as shown in Figure 1C. At different excitation wavelengths from 300 to 400 nm, the fluorescent spectra of the as-prepared PAH-Cu NCs varied (as shown in Figure 2A). The maximum fluorescence of the PAH-Cu NCs was emitted at 450 nm when the excitation wavelength was set at 360 nm (Figure 2B). When imaged by the high-resolution TEM, the particle size distribution of the PAH-Cu NCs was very uniform without aggregation (Figure 2C) in the range of 2.32 ± 0.53 nm. In the data of FT-IR spectrum (Figure 2D), N-H absorption bands at 1509 and 1603 cm^−1^ were both observed on PAH and the as-prepared PAH-Cu NCs. The peaks at 3036 and 3426 cm^−1^ originated from a broad NH^3+^ band and N-H stretching band in PAH [48], and both of the two peaks were observed on the as-prepared PAH-Cu NCs. According to the data of XPS (Figure 2E), the PAH-Cu NCs were composed of C1s, N1s, O1s, and Cu2p. The data demonstrated the binding of PAH and Cu NCs in the as-prepared PAH-Cu NCs. The XPS data were utilized to further evaluate the reduction type of copper in PAH-Cu NCs (Figure 2F). The two major peaks, i.e., 932.2 eV and 952.4 eV, which were stand for Cu 2p_3/2_ and Cu 2p_1/2_ presented in the binding energies of Cu 2p, respectively. The binding energy of 934 eV was not observed, indicating the absence of Cu^2+^ in the as-prepared Cu NCs [10,49]. However, Cu^+^ could not be excluded from the Cu NCs, because the binding energy at Cu 2p_3/2_ was only ~0.1 eV different from Cu [7]; therefore, supporting that the as-prepared PAH-Cu NCs were probably composed of Cu and Cu^+^.

To realize the composition of Cu NCs, the as-prepared PAH-Cu NCs were analyzed by matrix-assisted laser-desorption ionization mass spectrum (MALDI-MS). As shown in Appendix A, the major *m*/*z* peak was observed at 421.208 standing for [Cu_6_+K+H]^+^ adducts, and the second *m*/*z* peak at 381.204 originating from [Cu_6_]. On the basis of the results of the MALDI-TOF MS, the composition of six copper atoms of the as-prepared PAH-Cu NCs was supported, which was similar to the results from a previous study [5] that reported the amount of copper in Cu NCs showed fluorescence lower than 13.

### 2.3. Fluorescent Stability of PAH-Cu NCs

To evaluate the fluorescent stability of PAH-Cu NCs, the as-prepared PAH-Cu NCs were continuously exposed to excitation light for a period of time. The fluorescent intensity of the PAH-Cu NCs was recorded within 60 min, and this was repeated three times. The fluorescent intensity was reduced by about 7.3% within 10 min during the continuous exposure of excitation light, which supported that the decreased fluorescence within 10 min resulted from the ligand-induced fluorescence change [50]. The ligand binding with Cu NCs directly led to a change in initial fluorescence as soon as it was exposed to excitation light. However, the fluorescent intensity was stable between 10 and 60 min (Appendix A, about a decrease of 1.5% fluorescence), indicating highly stable fluorescence of PAH-Cu NCs during long-time exposure of excitation wavelength.

### 2.4. Selectivity of PAH-Cu NCs for Sensing DFX

To prevent the interference in plasma affecting sensing and to investigate the selectivity of PAH-Cu NCs for detecting DFX, several metal ions and amino acids, including Ca^2+^, Mg^2+^, K^+^, Na^+^, Zn^2+^, Ni^2+^, Pb^2+^, Co^2+^, Hg^2+^, Cd^2+^, Cr^3.+^, alanine, arginine, cysteine, glutamine, glycine, lysine, phenylalanine, and serine were tested for selectivity evaluation. All of the reagents for the selective test were used at a concentration of 100 μg/mL. Additionally, EDTA and glycerol, which are similar to DFX having multiple carboxylic groups or hydroxyl groups, were also used to test the selectivity of PAH-Cu NCs. The analytes did not all affect the fluorescent intensity of the PAH-Cu NCs for detecting DFX in standard solution, except for EDTA (Figure 3A). This was due to the fact that EDTA was able to chelate the copper of PAH-Cu NCs resulting in decreased fluorescence of PAH-Cu NCs. However, when the selectivity test was accomplished in the real plasma samples, there was good selectivity of PAH-Cu NCs for sensing DFX, because the real plasma samples were pretreated to prevent interference affecting the detection of DFX. After pretreatment of the plasma samples, only DFX showed a decrease in the fluorescent intensity of PAH-Cu NCs (Figure 3B). The data demonstrated that the PAH-Cu NCs had good selectivity for sensing DFX in plasma. In a previous study [39], DFX which could specifically bind with copper ions was demonstrated, although DFX was used for therapy of iron overloading in β-thalassemia patients. This explains why the DFX could specifically quench the fluorescence of the PAH-Cu NCs due to binding of the copper ions, and induce the aggregation of the PAH-Cu NCs (the data are shown in Figure 4A).

### 2.5. Validation of PAH-Cu NCs for Sensing DFX in Plasma

In this study, different concentrations of DFX spiked in plasma were used to react with PAH-Cu NCs to identify the fluorescent quenching effect between DFX and PAH-Cu NCs. The TEM images demonstrate the interaction between DFX and PAH-Cu NCs, as shown in Figure 4A, and a remarkable aggregation was observed after 50 µg/mL of DFX was reacted with PAH-Cu NCs. This supports that DFX can effectively chelate copper from PAH-Cu NCs. The loss of copper resulted in aggregation of the PAH-Cu NCs, and further decreased the fluorescent intensity. The aggregation of the PAH-Cu NCs could also be observed using the FT-IR data (Figure 4B). The absorption bands of 3036 and 3426 cm^−1^ belonging to broad NH^3+^ and the N-H band in PAH decreased by adding higher concentrations of DFX (Figure 4B). A similar mechanism was also found with EDTA, a common chelating agent. Therefore, EDTA can decrease the fluorescence of PAH-Cu NCs (Figure 3A). However, in plasma samples, these chelating agents that affect the detection of DFX using PAH-Cu NCs could be excluded through the pretreatment of samples (Figure 3B). The stable fluorescence of the PAH-Cu NCs could be specifically utilized for determination of DFX in plasma.

The calibration curve of DFX in this sensing method was established by spiking different concentrations of DFX into plasma. The regression equation of DFX is Y = (0.0173 ± 0.0001)X + (1.0166 ± 0.0123) with a suitable detection range for clinical practice and high linearity (r ≥0.9925). The linear range of DFX is from 1 μg/mL (2.68 μM) to 100 μg/mL (268 μM) (Figure 5). The detection limit of DFX is 0.1 μg/mL with a signal-to-noise ratio of three. The assays for precision and accuracy of the regression equation were investigated using three different concentrations of DFX (5, 15, and 75 μg/mL) spiked into plasma and detected by PAH-Cu NCs, for three replications, and calculated using relative standard deviation (RSD) and relative error (RE), respectively. The results are shown in Table 1. The RSD and RE values for the intra-day and inter-day assays were all between 1.40 and 6.51% and between 1.08 and 7.57%, respectively. The results indicated good precision and accuracy of the as-prepared PAH-Cu NCs for the detection of DFX in plasma.

### 2.6. Application for Real β-Thalassemia Patient Plasmas

The application of PAH-Cu NCs for real plasma samples from three patients who suffered β-thalassemia and were treated with DFX at the Kaohsiung Medical University Chung-Ho Memorial Hospital were completed in this study. The ethics approval was examined and approved by the Institutional Review Board at Kaohsiung Medical University Hospital. The participants all signed informed consents, after the content and risk of the study were explained. The plasma samples from the β-thalassemia patients were pretreated according to the procedure in Section 3.4 and detected by using PAH-Cu NCs, for three replications. Meanwhile, the patients’ DFX data detected using the CE method [38] which has been established by our lab were used to compare with the data using PAH-Cu NCs. Table 2 shows the concentrations of DFX for the three patients analyzed by the CE method and the PAH-Cu NCs. The concentrations of DFX by the CE method ranged from 15.33 to 33.63, and using PAH-Cu NCs the concentrations of DFX ranged from 15.76 to 37.24 µg/mL. In comparison of the data from the two methods showed that the relative error (RE) of the three samples was less than 10.73%, indicating the data obtained from PAH-Cu NCs was similar to our previous CE data [38]. The results supported that the PAH-Cu NC fluorescent method has potential for clinical application to sense DFX in β-thalassemia patients.

## 3. Materials and Methods

### 3.1. Materials

All chemicals and reagents were analytical grade and purchased from commercial suppliers. CuSO_4_ (≥99.0%), poly(allylamine hydrochloride) (PAH, Mw~17500), L-ascorbic acid (LAA), hydrochloric acid (HCl), amino acids, and EDTA were purchased from Sigma-Aldrich (St. Louis, MO, USA). DFX (98%) was purchased from Toronto Research Chemicals (North York, Canada). Other metal ions (Ca^2+^, Cd^2+^, Cr^3+^, Pb^2+^, Mg^2+^, K^+^, Na^+^, Ni^2+^, Zn^2+^, Hg^2+^, and Co^2+^) were kindly provided from Prof. Yen-Ling Chen at Kaohsiung Medical University. Methanol, acetonitrile (ACN), glycerol, and diisopropyl ether were obtained from E. Merck (Darmstadt, Germany). Ultrapure water was manufactured from a Millipore system and was used for synthesis of the PAH-Cu NCs and preparation of the reagents.

### 3.2. Apparatus

The fluorescent intensity of PAH-Cu NCs was measured by using a Hitachi F-4500 Fluorescence spectrometer (Tokyo, Japan). The excitation and emission wavelengths were set at 360 and 450 nm, respectively. The slits were set at 10 nm. The UV-visible spectra of each material were measured using a Hitachi U-5100 UV-Visible spectrophotometer (Tokyo, Japan). The pH value of buffer was adjusted using a DKK-TOA pH meter HM-25R (Tokyo, Japan). The X-ray photoelectron spectroscopy (XPS) was determined by ULVAC-PHI PHI 5000 VersaProbe. The transmission electron microscopy (TEM) images of PAH-Cu NCs were obtained by using a JEOL JEM-2100 transmission electron microscopy at a voltage of 200 kV. Fourier transform infrared spectroscopy (FT-IR) spectra of PAH-Cu NCs were measured using a Perkin-Elmer system 2000.

### 3.3. Synthesis of PAH-Cu NCs

CuSO_4_ (0.1 M), PAH (10%), and LAA (1.0 M) were all dissolved in ultrapure water. A total of 20 mL solution containing CuSO_4_ (0.2 mL, 0.1 M) and PAH (0.1 mL, 10%) were mixed with water and stirred for 2 min. Then, LAA (0.2 mL, 1 M) was added to the solution and gently stirred at 60 °C for 2 h. When the LAA was added into solution, the color of solution changed from yellowish to chrome yellow. The as-prepared PAH-Cu NCs solution was stored at 4 °C until analysis.

### 3.4. Sample Pretreatment and Sensing

The stock solution of DFX was prepared in methanol (5 mg/mL) and stored at −20 °C, and then diluted to identical concentrations using the plasma from healthy volunteers. Plasma (200 μL, with/without DFX) was deproteinized by 400 μL ACN and centrifuged at 12,000 rpm for 2 min. After centrifugation, 550 μL supernatant was reclaimed and dried in vacuum for 2 h at 40 °C. The residues were redissolved in 20 μL HCl (0.1 M), and then 500 μL diisopropyl ether was added into the solution to perform the liquid–liquid extraction. After vigorous shaking and centrifugation, 950 μL organic solvent was reclaimed and dried again for 5–10 min at room temperature. The dry residual was reconstituted with a solution containing 50 μL methanol and 150 μL PAH-Cu NCs, and then equilibrated for 5 min. Finally, the fluorescence of the solution was measured using the fluorescent spectrometer (λ_ex_/λ_em_ = 360/450 nm).

### 3.5. Selectivity for Measurement of DFX

The selectivity of the PAH-Cu NCs was evaluated using various essential trace elements and amino acids in the human body, including Ca^2+^, Mg^2+^, K^+^, Na^+^, Zn^2+^, Ni^2+^, Pb^2+^, Co^2+^, Hg^2+^, Cd^2+^, Cr^3.+^, alanine, arginine, cysteine, glutamine, glycine, lysine, phenylalanine, and serine. EDTA and glycerol were also used to evaluate the interaction of PAH-Cu NCs. In experimental practice, all the reagents for the selective testing were used in the concentration of 100 μg/mL.

### 3.6. Determination of DFX in Real Samples

The real plasma samples were recruited from β-thalassemia patients, who had been taking DFX regularly for more than a year, in the department of Pediatrics, Kaohsiung Medical University hospital. The recruited healthy volunteers and patients were all informed about the experiment and signed consent forms. The real plasma samples were obtained from the β-thalassemia patients, following the sample pretreatment procedure described in Section 3.4.

## 4. Conclusions

To prevent the adverse effects and evaluate the efficacy of DFX, development of a method for fast and easy detection of the concentration of DFX is useful and absolutely necessary. In this study, a facile and environmentally friendly sensor, PAH-Cu NCs, was developed for monitoring DFX in the plasma of β-thalassemia patients. The PAH-Cu NCs showed rapid reaction and good response to DFX, and exhibited good selectivity for distinguishing deferoxamine, deferiprone, and other chelating agents after pretreatment of the plasma samples. The calibration curve was also shown an excellent linearity for detecting DFX in plasma. When compared with previous methods for determination of DFX, this PAH-Cu NCs also exhibited the good linear range (Table 3). Furthermore, this PAH-Cu NCs fluorescent method was successfully applied for clinical β-thalassemia patients, and the results were similar with that obtained by using previous CE method which was developed by our lab. Additionally, the fluorescence of PAH-Cu NCs was stable for the long time of exposing excitation wavelength indicating the high stability of the fluorescence of PAH-Cu NCs. All of the data demonstrated the applicability of this PAH-Cu NCs fluorescent sensor for determination of DFX in clinical, and the simple and novel sensors could be served as a tool for a clinical survey.

## Data Availability

Data is contained within the article or Appendix A.

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
