# Peer review of "Stable Luminescent Poly(Allylaminehydrochloride)-Templated Copper Nanoclusters for Selectively Turn-Off Sensing of Deferasirox in β-Thalassemia Plasma"

_pharmaceuticals, 2021, doi:10.3390/ph14121314_

Round 1

Reviewer 1 Report

The manuscript "A Stable Luminescent Poly(allylaminehydrochloride)-Templated Copper Nanoclusters for Selectively Turn-Off Sensing of Defer-Asirox in β-Thalassemia Plasma" by Hung-Ju Lin, Chun-Chi Wang, Hwang-Shang Kou, Cheng-Wei Cheng and Shou-Mei Wu contains a study where copper nanoparticles/clusters are first synthesized, characterized, and then are used in the sensing of a biologically relevant molecule. The study contains some interesting results, and the authors have performed a robust assessment of the synthetic conditions, but there are many unclear areas. In addition, there are questions on the validity of some reasoning and results. These points must be addressed before the work is published. My comments and criticisms follow below.

1. Perhaps the most important point is why was copper chosen as a probe nanoparticle material in this work? While it is true that copper possess some positive properties, it is notoriously easy to oxidize. Gold nanoparticles on the other hand are immune to oxidation and also show fluorescence activity, they would be much easier to handle as compared to copper. The vast majority of studies with copper nanomaterials must take special care to avoid oxidation, but the point is not addressed in this work. This also has implications to data presented in the manuscript.

2. Copper nanoparticles possess surface plasmon resonance, which makes them a candidate for use with UV-Visible spectroscopy in biomolecule sensing applications. UV-Vis is much easier to employ than fluorescence, so why was UV-Vis not used as the detection technique in this study? At least UV-Vis should be used to characterize the nanoparticle materials at different points in the study.

3. Copper nanoparticles seem to be prepared in ambient atmosphere, which will lead to oxidation in a very fast timeframe (irrespective of reaction conditions). Were any techniques used to suppress oxidation (such as using an argon atmosphere?). In addition, the particles oxidation and stability will affect sensing measurements, how long were the particles kept before being used in subsequent experiments?

4. Selectivity experiments for the copper particle material is suspect. The addition of nearly any of these materials will cause the particles to aggregate and/or oxidize. The key may be the level of salt concentration present (assists both processes). In any event, some experiments use purified blood plasma, which contains much more than the target analyte. While the experiments suggest sensitivity to DFX, its possible any of the other plasma contents is responsible. Without controlling the actual plasma contents, these results are very uncertain.

5. Figure 2C shows a TEM image of the copper nanomaterial. Relatively small copper particles seem to be observed in the image, but there is some large, spherical feature in the upper right. Which material is expected? and how do you know you have copper clusters? Typically UV-Vis and/or XRD analysis would be used to study this. 

6. Figure 2E,F shows the XPS results, and as stated in the text cannot differentiate between metallic copper or copper oxide. In this case, some other technique should be used to make the assignment. The point may seem small, but without  confirmation of what the probe material is composed of, the experimental results cannot be fully interpreted or reproduced. 

7. Figure 4A is a TEM image that causes many questions and concerns. This is an image of the copper material after interaction with DFX. This image shows the telltale signs of copper nanoparticle oxidation, aggregation and coalescence. The effect on the material optical properties would be severe. This result shows that too many characteristics of the copper material have been changed for a reliable analysis. In a typical sensing application, only one thing should change, such as particle aggregation. Whatever is happening to the particles in this case cannot be controlled, which makes the results in this study non-controllable. This once again raises the issue for why is copper used when more stable materials will give better results?

Author Response

  1. Perhaps the most important point is why was copper chosen as a probe nanoparticle material in this work? While it is true that copper possess some positive properties, it is notoriously easy to oxidize. Gold nanoparticles on the other hand are immune to oxidation and also show fluorescence activity, they would be much easier to handle as compared to copper. The vast majority of studies with copper nanomaterials must take special care to avoid oxidation, but the point is not addressed in this work. This also has implications to data presented in the manuscript.

Ans: Thanks for the reviewer’s comment. We have added the description in the text as shown in line 262-267 and line 272-278.  

Line 262-267:

Although the Cu NCs without templates for covering were easily oxidized and lost their fluorescence, they are cheaper and more easily for preparation than other nanomaterials, such as gold nanoparticles. Therefore, a lot of ligands were developed to cover the Cu NCs to avoid the oxidation of single Cu NCs [3, 8-17]. In this research, the polymer of PA was utilized as the ligands for covering the Cu NCs to avoid the oxidation. 

Line 272-278:

The ligand binding with Cu NCs directly leads to a change in initial fluorescence as soon as exposed with excitation light. However, the fluorescent intensity was stable between 10 min to 60 min (Fig. S4, about decrease of 1.5% fluorescence), indicating the highly stable fluorescence of PAH-Cu NCs for the long-time exposure of excitation wavelength. The data also demonstrate the use of the polymer, PAH, could effec-tively protect the Cu NCs to oxidation.

  1. Copper nanoparticles possess surface plasmon resonance, which makes them a candidate for use with UV-Visible spectroscopy in biomolecule sensing applications. UV-Vis is much easier to employ than fluorescence, so why was UV-Vis not used as the detection technique in this study? At least UV-Vis should be used to characterize the nanoparticle materials at different points in the study.

Ans: Thanks for the reviewer’s comment. We have added the UV-Vis spectrum of the PAH-Cu NCs as shown in the Fig. S3 and the description was as shown in line 258-259.

Fig. S3. The UV-Vis spectrum of the PAH-Cu NCs.

Line 258-259:

Additionally, the UV-Vis spectrum of the PAH-Cu NCs was as shown in Fig. S3.

  1. Copper nanoparticles seem to be prepared in ambient atmosphere, which will lead to oxidation in a very fast timeframe (irrespective of reaction conditions). Were any techniques used to suppress oxidation (such as using an argon atmosphere?). In addition, the particles oxidation and stability will affect sensing measurements, how long were the particles kept before being used in subsequent experiments?

Ans: Thanks for the reviewer’s comment. We have added the description in the text as shown in line 267-278.

Line 267-278:

To evaluate the fluorescent stability of PAH-Cu NCs, as-prepared PAH-Cu NCs was continuously exposed with excitation light for a period. The fluorescent intensity of PAH-Cu NCs was recorded within 60 minutes, and repeated for three times. The fluorescent intensity was reduced about 7.3% within 10 min in the continuous exposure of excitation light. That was supported the decrease of the fluorescence within 10 min was resulted from ligand-induced fluorescence change [50]. The ligand binding with Cu NCs directly leads to a change in initial fluorescence as soon as exposed with excitation light. However, the fluorescent intensity was stable between 10 min to 60 min (Fig. S3, about decrease of 1.5% fluorescence), indicating the highly stable fluorescence of PAH-Cu NCs for the long-time exposure of excitation wavelength. The data also demonstrate the use of the polymer, PAH, could effectively protect the Cu NCs to oxidation.

  1. Selectivity experiments for the copper particle material is suspect. The addition of nearly any of these materials will cause the particles to aggregate and/or oxidize. The key may be the level of salt concentration present (assists both processes). In any event, some experiments use purified blood plasma, which contains much more than the target analyte. While the experiments suggest sensitivity to DFX, its possible any of the other plasma contents is responsible. Without controlling the actual plasma contents, these results are very uncertain.

Ans: Thanks for the reviewer’s comment. The selectivity tests and calibration curves were established by using real plasma matrix spiked with DFX and the related description was as shown in line 291-298, line 305-308 and line 322-323.

Line 291-298:

However, when the selectivity test was accomplished in the real plasma samples, the selectivity of PAH-Cu NCs for sensing DFX become good. That was because the real plasma samples should be pretreated to prevent the interference affecting the detection of DFX. After pretreatment of the plasma samples, only DFX showed the decrease of the fluorescent intensity of PAH-Cu NCs (Fig. 3B). Additionally, the presence of high salt concentration in plasma would not affect the detection of DFX. The data demonstrated the PAH-Cu NCs had a good selectivity for sensing DFX in plasma.

Line 305-308:

In this study, all of the experiment was performed by using plasma matrix spiked with DFX. The different concentrations of DFX spiked in plasma were used to react with PAH-Cu NCs to identify the fluorescent quenching effect between DFX and PAH-Cu NCs. 

Line 322-323:

Calibration curve of DFX in this sensing method was established by spiking different concentrations of DFX into plasma.

  1. Figure 2C shows a TEM image of the copper nanomaterial. Relatively small copper particles seem to be observed in the image, but there is some large, spherical feature in the upper right. Which material is expected? and how do you know you have copper clusters? Typically UV-Vis and/or XRD analysis would be used to study this.

Ans: Thanks for the reviewer’s comment. We have added the UV-Vis spectrum of the PAH-Cu NCs as shown in the Fig. S3 and the description was as shown in line 258-259.

Fig. S3. The UV-Vis spectrum of the PAH-Cu NCs.

Line 258-259:

Additionally, the UV-Vis spectrum of the PAH-Cu NCs was as shown in Fig. S3.

  1. Figure 2E,F shows the XPS results, and as stated in the text cannot differentiate between metallic copper or copper oxide. In this case, some other technique should be used to make the assignment. The point may seem small, but without confirmation of what the probe material is composed of, the experimental results cannot be fully interpreted or reproduced.

Ans: Thanks for the reviewer’s comment. We have described that in the text as shown in line 239-247.

Line 239-247.

The XPS data was utilized to further evaluate the reduction type of copper in PAH-Cu NCs (Fig. 2F). The two major peaks, 932.2 eV and 952.4 eV, which were stand for Cu 2p3/2 and Cu 2p1/2 presented in the binding energies of Cu 2p, respective-ly. The binding energy of 934 eV was not observed, indicating the absence of Cu2+ in as-prepared Cu NCs [10,49]. However, Cu+ could not be excluded from Cu NCs, be-cause the binding energy at Cu 2p3/2 was only ~0.1 eV different from Cu [7]. There-fore, that was supported as-prepared PAH-Cu NCs probably composed of Cu and Cu+. From the XPS data of Cu 2p, the oxidation of the Cu NCs would not be found in the PAH-Cu NCs.

  1. Figure 4A is a TEM image that causes many questions and concerns. This is an image of the copper material after interaction with DFX. This image shows the telltale signs of copper nanoparticle oxidation, aggregation and coalescence. The effect on the material optical properties would be severe. This result shows that too many characteristics of the copper material have been changed for a reliable analysis. In a typical sensing application, only one thing should change, such as particle aggregation. Whatever is happening to the particles in this case cannot be controlled, which makes the results in this study non-controllable. This once again raises the issue for why is copper used when more stable materials will give better results?

Ans: Thanks for the reviewer’s comment. After spiking the DFX into plasma, the aggregation of the PAH-Cu NCs indeed happened in the presence of DFX. We have described the situation in the text as shown in line 308-316.

Line 308-316.

The image of TEM demonstrated the interaction between DFX and PAH-Cu NCs as shown in Fig. 4A, and a remarkable aggregation was observed after 50 µg/ml DFX was reacted with PAH-Cu NCs. That was supported the DFX has a good ability to chelating the copper from the PAH-Cu NCs. The loss of the copper would result in the aggregation of the PAH-Cu NCs, and further the decrease of the fluorescent intensity. The aggregation of the PAH-Cu NCs could also be observed by the data of FT-IR (Fig. 4B). The absorption bands of 3036 cm-1 and 3426 cm-1 belonging to broad NH3+ and N-H band in PAH decreased by adding higher concentrations of DFX (Fig. 4B).

Reviewer 2 Report

1. Please show the equation how to calculate the RSD and RE.
2. The conditions of MALDI-TOF MS are missing.
3. Fig. S2 caption, the term "MALDI-MS spectrum" is unsuitable, please use "MALDI-TOF MS spectrum". 
4. Section 2.1. Materials, metal ions should be mentioned the name of commercial company. The sentence "Other metal ions were kindly provided from..." is unsuitable.
5. Do authors test the recovery of the proposed method?

Author Response

  1. Please show the equation how to calculate the RSD and RE.

Ans: Thanks for your kind suggestion. We have done the modification in the text as shown in line 327-331 and 336-346. .

Line 327-331:

Precision and accuracy assay of this regression equation were investigated by three different concentrations of DFX (5, 15, 75 μg/ml) spiked into plasma and detected by the PAH-Cu NCs for three times, and evaluated using relative standard deviation (RSD) and relative error (RE), respectively. RSD and RE were calculated by equation (1) and (2).

Line 336-346:

RSD = (S / X)*100%  Equation (1)

RSD = relative standard deviation
S = standard deviation
X = mean of the data.

RE = [(V measured – Vactual) / Vactual] *100%  Equation (2)

RE = relative error
V measured = measured value of data
Vactual = actual value of data

  1. The conditions of MALDI-TOF MS are missing.

Ans: Thanks for your kind suggestion. We have done the modification in the text as shown in line 248-255.

Line 248-255:

To realize the composition of Cu NCs, the as-prepared PAH-Cu NCs were analyzed by matrix-assisted laser desorption ionization-time of flight mass spectrometry (MALDI-TOF MS). This spectrum data (Fig. S3) was obtained by using a 5800 MALDI TOF mass spectrometer (AB SCIEX, Framingham, MA, USA) in positive mode with trans-2-[3-(4-tert-butylphenyl)-2-methyl-2-propenylene]malononitrile (DCTB) as a matrix. As shown in Fig. S2, the major m/z peak was observed at 421.208 standing for [Cu6+K+H]+ adducts, and the second m/z peak at 381.204 originating from [Cu6].

  1. Fig. S2 caption, the term "MALDI-MS spectrum" is unsuitable, please use "MALDI-TOF MS spectrum".

Ans: Thanks for your kind suggestion. We have modified the Fig. S2 caption, and it has been shown in the text.

Fig. S2. The MALDI-TOF MS spectrum of PAH-Cu NCs. This spectrum data was obtained by using a 5800 MALDI TOF mass spectrometer (AB SCIEX, Framingham, MA, USA) in positive mode with trans-2-[3-(4-tert-butylphenyl)-2-methyl-2-propenylene]malononitrile (DCTB) as a matrix.

  1. Section 2.1. Materials, metal ions should be mentioned the name of commercial company. The sentence "Other metal ions were kindly provided from..." is unsuitable.

Ans: Thanks for your kind suggestion. We have done the modification as shown in the Section 2.1. Materials.

Other metal ions (Ca2+, Cd2+, Cr3+, Pb2+, Mg2+, K+, Na+, Ni2+, Zn2+, Hg2+, and Co2+ prepared from their chloride salts) were purchased from Sigma-Aldrich (St. Louis, MO USA).

  1. Do authors test the recovery of the proposed method?

Ans: Thanks for your kind suggestion. We have done the modification as shown in Table 1.

Table 1. Precision and accuracy for the determination of DFX in intra-day and inter-day analysis.

Concentration spiked (μg/mL)

Concentration detected (μg/mL)

RSD (%)

RE (%)

Recovery (%)

Intra-day (n=3)

 DFX

5.00

5.38 (±0.34)

6.38

7.57

107.57

15.00

15.91 (±0.40)

2.52

6.07

106.07

75.00

73.60 (±1.87)

2.54

-1.87

98.13

Inter-day (n=5)

 DFX

5.00

4.85 (±0.32)

6.51

-2.98

97.02

15.00

15.16 (±0.97)

6.41

1.08

101.08

75.00

76.81 (±1.08)

1.40

2.42

102.42

Round 2

Reviewer 1 Report

The authors have made extensive changes to the manuscript and have added more description and arguments in previously unclear points. While I do not think that the authors completely addressed my concerns (most importantly about the state of the copper nanoparticle material before and after the detection), it may be appropriate to allow readers of the article to judge for themselves.